# A High-Throughput Method for Quantifying *Drosophila* Fecundity

**DOI:** 10.3390/toxics12090658

**Published:** 2024-09-09

**Authors:** Andreana Gomez, Sergio Gonzalez, Ashwini Oke, Jiayu Luo, Johnny B. Duong, Raymond M. Esquerra, Thomas Zimmerman, Sara Capponi, Jennifer C. Fung, Todd G. Nystul

**Affiliations:** 1Department of Anatomy, University of California, San Francisco, CA 94143, USA; 2Department of Biology, San Francisco State University, San Francisco, CA 94132, USA; 3Center for Cellular Construction, San Francisco, CA 94158, USA; 4OB/GYN Department, University of California, San Francisco, CA 94143, USA; 5Center for Reproductive Sciences, University of California, San Francisco, CA 94143, USA; 6Department of Chemistry and Biochemistry, San Francisco State University, San Francisco, CA 94132, USA; 7IBM Almaden Research Center, San Jose, CA 95120, USA; 8San Francisco EaRTH Center, University of California, San Francisco, CA 94143, USA

**Keywords:** reproductive toxicology, *Drosophila*, oogenesis

## Abstract

The fruit fly, *Drosophila melanogaster*, is an experimentally tractable model system that has recently emerged as a powerful “new approach methodology” (NAM) for chemical safety testing. As oogenesis is well conserved at the molecular and cellular level, measurements of *Drosophila* fecundity can be useful for identifying chemicals that affect reproductive health across species. However, standard *Drosophila* fecundity assays have been difficult to perform in a high-throughput manner because experimental factors such as the physiological state of the flies and environmental cues must be carefully controlled to achieve consistent results. In addition, exposing flies to a large number of different experimental conditions (such as chemical additives in the diet) and manually counting the number of eggs laid to determine the impact on fecundity is time-consuming. We have overcome these challenges by combining a new multiwell fly culture strategy with a novel 3D-printed fly transfer device to rapidly and accurately transfer flies from one plate to another, the RoboCam, a low-cost, custom-built robotic camera to capture images of the wells automatically, and an image segmentation pipeline to automatically identify and quantify eggs. We show that this method is compatible with robust and consistent egg laying throughout the assay period and demonstrate that the automated pipeline for quantifying fecundity is very accurate (r^2^ = 0.98 for the correlation between the automated egg counts and the ground truth). In addition, we show that this method can be used to efficiently detect the effects on fecundity induced by dietary exposure to chemicals. Taken together, this strategy substantially increases the efficiency and reproducibility of high-throughput egg-laying assays that require exposing flies to multiple different media conditions.

## 1. Introduction

With the rapid rise of chemical pollution in the environment [1], there is an urgent need for improved strategies to identify chemicals that pose a risk to human health and the ecosystem. To address this need, “new approach methodologies” (NAMs) are being developed that allow for high-throughput screening of potentially harmful chemicals without the use of mammalian model organisms [2,3,4]. There is an increasing interest in the use of *Drosophila melanogaster* screens as NAMs, as they are relatively inexpensive, avoid the ethical concerns associated with mammalian models, utilize convenient and efficient modes of chemical exposure, and lay the groundwork for follow-up studies that can take advantage of the wealth of experimental tools available for *Drosophila* research [5,6,7,8]. As *Drosophila* is an excellent model for oogenesis [9], we sought to develop a method that would facilitate the use of *Drosophila* fecundity assays as a tool to identify chemicals that pose a risk to reproductive health.

Measurements of *Drosophila* fecundity are used in a wide variety of studies in addition to toxicology, including investigations of aging, stem cell biology, nutrition, and behavior [10,11,12,13,14,15,16]. These studies build on the wealth of knowledge about *Drosophila* oogenesis and the complex array of inputs that combine to optimize the rate of egg laying in response to the environment. Environmental cues are sensed primarily through pheromones [17,18,19], which are detected by the olfactory system, and nutrient cues, which are received through the digestive tract and sensed by the fat body [20,21,22]. These upstream signals coordinate the release of hormonal signals into the hemolymph that regulate the cell- and tissue-level responses throughout the body. For example, within the ovary, these signals act on both the stem cell compartment, called the germarium, to regulate the rate of cell division [23,24,25], as well as on later stages of oogenesis to promote follicle survival and maturation [26,27,28]. Intercellular signals also trigger ovulation of mature eggs into the uterus, fertilization, and egg deposition [29,30,31,32]. Thus, measurements of fecundity provide a real-time, quantitative assessment of this multiorgan process in live flies. This makes the assay suitable for longitudinal studies such as investigating how fecundity changes with age or with prolonged chemical exposure. However, as the culmination of a broad cascade of inputs, fecundity assays must be carefully designed to ensure robust and reproducible results.

Robust egg laying occurs during the first four weeks of adult life [10]. Environmental conditions that promote a high, consistent rate of egg laying include the presence of males, a lack of overcrowding, room temperature, and a protein-rich diet, which is sometimes provided by the addition of a wet yeast paste to culture vials with standard media [20,33]. The fecundity rates of wildtype flies vary considerably between studies, ranging from 20 to 100 eggs per female per day [34,35]. A common approach to quantifying the rate of egg laying is to maintain flies in standard culture vials or molasses agar plates for a defined period of time and then to manually count the number of eggs that are visible under a dissecting microscope [32,34,36]. This method is simple and has the advantage that flies are maintained in standard conditions, but it is time-consuming and prone to variation due to human error across observations. In addition, this approach may underestimate the rate of egg laying because agar-based culture media and yeast paste are soft, so eggs can sink below the surface and become obscured from view. 

Therefore, several studies have modified this approach to address these challenges. For example, using a firmer agarose-based culture media is useful for keeping eggs on the surface [37], though eggs deposited directly in the yeast paste (if it is added) are still hard to see. To increase speed and improve reproducibility, several studies have developed methods to automate the process of identifying and counting eggs. These approaches start with images of eggs laid in a standard culture vial or large plate and apply an image segmentation algorithm to automatically identify and quantify the eggs [38,39,40]. Lastly, other studies have used multiwell plates rather than culture vials to increase the efficiency and reduce the cost of testing many different diets or chemical exposures [41,42,43,44]. However, a major drawback to the multiwell format is the difficulty of ensuring that flies remain in the same wells throughout the assay and of accurately transferring flies from one plate to another, as is typically needed for a multi-day time course. Each of these innovations is useful on its own but they have been optimized for a particular application and thus are not easily combined into a single workflow. 

Here, we describe a strategy for performing a high throughput multi-day fecundity assay and demonstrate that it is effective for measuring the impact of chemical additives in the diet. Building on previous innovations, our strategy combines a multiwell format with a novel 3D-printed fly transfer device to rapidly and accurately transfer flies from one plate to another, the RoboCam, a low-cost, custom-built robotic camera to automatically capture images of the wells, and an image segmentation pipeline to automatically identify and quantify eggs. We show that this method is compatible with robust and consistent egg laying throughout the assay period and that it can accurately detect chemical-induced effects on fecundity. Taken together, this strategy substantially increases the efficiency and reproducibility of high-throughput egg-laying assays that require exposing flies to multiple different media conditions. 

## 2. Materials and Methods

### 2.1. Culturing Conditions and Fecundity Assay

The egg-laying medium was made by diluting 100% Concord Grape Juice (Santa Cruz Organic) in water to a final concentration of 20% grape juice and adding sucrose (Sigma-Aldrich Saint Louis, MO, USA, Cat# 84100) at 0.1 g/mL and agarose powder (Genesee Scientific, El Cajon, CA, USA, Cat# 20-101) at either 0.01 g/mL or 0.02 g/mL (to achieve an agarose concentration of 1% or 2%, respectively). The solution was boiled to melt the agarose powder, and exactly 300 µL of the hot solution was added to each well, with care taken to minimize bubbles and inaccurate pipetting due to the viscous nature of the media. Being precise with the pipetting ensures that the surface of the media is the same height in each well, which is important for obtaining clear images with the RoboCam. Once cooled, 30 µL of a wet yeast slurry (0.2 g/mL of Red Star yeast granules dissolved in water) was added on top of the media, and the sides of the plate were tapped until the solution evenly covered the entire surface of the gel. The plates were then dried in a fume hood until the yeast layer was completely dry (approximately 90 min). This was important to ensure that the flies did not get stuck in the media.

Flies from the *Drosophila melanogaster* strain *w1118* (BDSC Stock # 3605) were raised on standard molasses food at 25 °C. The bottles were cleared of adults and then newly eclosed flies were fed wet yeast daily for one week before starting the fecundity assays. Flies were then maintained on an egg-laying medium with a layer of wet yeast on top in 48-well cell culture plates (Genesee Scientific, 25−103) for a 24 h “pre-egg-laying period”. This provided time for the small cohorts of flies in each well to become accustomed to the new environment and receive a uniform nutritional experience before the fecundity measurements began. 

After the pre-egg-laying period, the flies were maintained on the egg-laying medium with yeast on top as described above, either without any other additives, with DMSO, or with a chemical dissolved DMSO, as described in the “Preparation of chemical solutions” section below. Flies were transferred to new plates containing fresh medium with yeast on top every day during the days when quantification of fecundity was not required. On days when quantification of fecundity was required, the yeast slurry was added to the egg-laying medium while it was still warm, and the solution was mixed to homogeneity before pipetting into the wells. This modification was important to provide an optically uniform surface for egg laying while still maintaining a protein-rich diet.

### 2.2. Construction of the 3D-Printed Fly Transfer Device

The fly transfer lid was constructed from a sheet of 0.6 mm nylon mesh mounted between a lower array of cups (8 mm diameter, 10 mm deep, 6-degree taper) and an upper array of gas exchange holes. The lower and upper layers were designed with Tinkercad (www.tinkercad.com, accessed on 6 August 2024) and stereo lithographically 3D printed with LEDO 6060 resin (www.jlcpcb.com, accessed on 6 August 2024). Studs (3 mm tall, 2.5 mm diameter) in the upper layer fit into matching holes (3 mm diameter) in the nylon sheet and lower layer, sealed together with cyanoacrylate glue. The holes in the nylon mesh were formed by melting the desired nylon hole locations with a hot soldering iron guided by a metal template. 

### 2.3. Adding and Transferring Flies Using a 3D-Printed Fly Transfer Device 

A 3D-printed transfer device was used to keep the flies in individual wells during the culture period and ensure accurate transfer to new wells. The transfer device was placed on a CO_2_ pad and two females and one male were added into each cylinder of the transfer device. The plate containing the egg-laying medium was placed on top of the lid and secured to ensure proper fit into each well. The plate and transfer device were removed from the CO_2_ pad and flipped over so that the flies landed on the media. When the flies recovered and began flying, the plate, with the transfer device as a lid, was placed in a 25 °C incubator. To transfer flies from one plate to another, the plate and transfer device were placed mesh-side down on a CO_2_ pad. After the flies dropped to the mesh lining of the lid, the plate was removed and replaced with a new plate. The older plate was either saved for imaging or discarded. See Appendix A.

### 2.4. Construction of the RoboCam

To build the RoboCam, a high-resolution digital camera (12 MP HQ Camera, Adafruit, New York, NY, USA) equipped with a 16 mm telephoto lens (Adafruit, New York, NY, USA) was attached to the head of a conventional 3D printer (AnyCubic Mega S, Shenzhen, China) and an LED Light Tracing Box was placed on the 3D printer plate so that it illuminated them from below (Appendix A). A single-board Raspberry Pi computer (Adafruit) controlled the x, y, and z camera location using G-code commands. G-code is a standard programming language for 3D printers that provides instructions to the machine on where to move, how fast to move, and which path to follow. The G-code commands were generated by the user with a Graphical User Interface (GUI) running on the Raspberry Pi, written in Python with PySimpleGUI and OpenCV libraries. The well images collected by the RoboCam were stored on an external hard drive. See Appendix A for detailed descriptions of how to build and operate a RoboCam.

### 2.5. Imaging

To collect egg-laying data, flies were removed exactly 24 h after placement into the wells and the plate was either placed in a −20 °C freezer for imaging later (no more than 48 h) or imaged immediately. For imaging, the plate was placed on the RoboCam platform; the camera position was registered to the wells in each of the four corners; the z-focus was adjusted as needed; and the movement and acquisition module was initiated. This module moves the camera to a precise position over each well and takes a single image in the specified z-position. When the z-position was not optimal for all of the wells on the plate, the module was run additional times at different z-positions, as needed. 

### 2.6. Imaging Pipeline Implementation

The imaging pipeline was written in Python and provided as a Google CoLab notebook. To run the imaging pipeline, the necessary input and output file paths and the estimated well diameter were provided as inputs. Then, Stardist [45,46] was installed (https://github.com/stardist/stardist, accessed on 6 August 2024), the necessary libraries were loaded, several functions were defined, and the imaging processing steps were performed. Briefly, the image processing steps were (1) preprocessing, which included inverting the image, converting it to 8-bit grey scale, and normalizing the pixel intensity; (2) well detection using the canny edge detection module from the Python package, skimage; (3) image segmentation using flyModel2 (the default probability threshold of 0.7 can be manually adjusted based on the quality of the images) to identify eggs; (4) a filtering step, in which segmented objects that were outside of the predicted well, less than 2500 pixels^2^ or greater than 4000 pixels^2^, were excluded; and (5) an output step in which the number of objects (eggs) per well were exported as an Excel file and images showing the segmentation results were saved. 

### 2.7. Quantifly Analysis

Quantifly [40] was trained using 19 images according to the instructions provided by the manual. The software was trained at different values of sigma. Sigma value of 2 gave the most accurate counts and was used for comparison with the Stardist models. 

### 2.8. Stardist Model Training

Fifty-two randomly chosen images were used for annotation using Labkit plugin in Fiji. The resulting pairs of images and masks were used to train a Stardist model by adapting the example Python notebook (https://github.com/stardist/stardist/blob/master/examples/2D/2_training.ipynb, accessed on 6 August 2024) for this purpose. Data augmentation and training were performed as suggested with a split of 44 training images and 8 validation images. The optimized probability threshold for flyModel2 is 0.706 and the nms_thresh is 0.3. The key metrics for flyModel2 at t = 0.5 are listed as follows: fp = 3; tp = 50; fn = 2; precision = 0.9434; recall = 0.9615; accuracy = 0.909. 

### 2.9. Preparation of Chemical Solutions

Briefly, 1000× stock solutions in 100% DMSO (Sigma-Aldrich, D8418) were diluted 1:100 with water to achieve a 10× concentration of chemical in 1% DMSO. Then, 30 µL of the 10× chemical solution was added into each well to achieve a final concentration of 1x chemical in 0.1% DMSO. Specifically, to achieve 0.1 µM, 10 µM, and 25 µM final concentrations of rapamycin (bioWORLD, 41810000-2) and bendiocarb (Sigma-Aldrich, 45336), the 1000× stock solutions were 0.1 mM, 10 mM, and 25 mM, respectively. 

### 2.10. Chemical Exposure Time Course

To quantify the effect of chemical exposure on fecundity and viability, flies were maintained in the 48-well plates with either 0.1% DMSO alone or 0.1% DMSO with chemical (to create the control and experimental conditions, respectively), with transfer to a new plate every day for 7 days. The egg-laying medium with the yeast mixed and 0.1% DMSO with or without chemical in was used on days 1, 3, and 7 so that fecundity could be quantified, and egg-laying medium with the yeast on top and 0.1% DMSO with or without chemical was used on all other days. To thoroughly assess reproducibility, each chemical condition was repeated in at least 16 wells per replicate, and data are an aggregate of seven replicates. Every time point of every condition was not included in every replicate, but there were at least 60 wells total across all replicates for every time point of every condition (see Appendix A. Rmd in the Github repository for more details). Fecundity data were not collected from any condition in which one or more females did not survive in at least half of the wells. When both females survived in at least half of the wells, egg-laying counts were performed on all wells in which both females were alive at the time of transfer to a new plate. 

### 2.11. Data and Code Availability

All raw data and software code are publicly available on the Nystul Lab Github site at https://github.com/NystulLab/HighThroughputFecundityAssay (accessed on 6 August 2024). RoboCam videos can also be viewed at https://www.youtube.com/channel/UCa-Fm65VvaGldKf60zEHuVw (accessed on 6 August 2024).

## 3. Results

### 3.1. Multiwell Culture Conditions

The first step in creating a high-throughput assay for quantifying fecundity was to develop a procedure for culturing flies in a 48-well format that promoted robust, reproducible egg laying onto a surface that could be clearly imaged and is compatible with controlled exposure to chemicals or other additives to the diet. We found that a standard grape juice and agarose recipe is a good base, as it is firm enough to prevent eggs from sinking below the surface, optically uniform, and resistant to drying and cracking over a 24 h period. However, the typical procedure of adding a dollop of yeast paste to the side of the chamber is too cumbersome and imprecise for the high-throughput 48-well format. In addition, with the yeast paste physically separated from the base media, the nutritional experiences and exposures to chemicals in the diet are more variable as they depend on the amount that individual flies consume from each nutritional source. Mixing dried yeast into the grape juice and agarose mixture before it hardened is more efficient and ensures a uniform nutritional experience. However, while this method supports robust egg laying for 24 h, we found that fecundity drops significantly when flies are maintained on this media for more than one day (Figure 1). This is likely due to an absence of wet yeast paste, rather than the composition of the media, as similar results were obtained when we used the standard molasses or BDSC recipes (Figure 1). 

We next investigated a third option, which was to apply a defined amount of a wet yeast slurry (created by adding more water than would be used to make yeast paste) to the surface of the media with a pipetman and then allow the excess water to evaporate from the plates so that a moist, even layer of yeast coated the entire surface of the media. This created a uniform dietary exposure and supported robust egg laying but we found that it is difficult to reliably detect all the eggs laid in this condition because the yeast layer is soft and not optically uniform.

Therefore, we devised a protocol that took advantage of all three of these methods (Figure 2A). First, a large cohort of freshly eclosed flies is maintained in standard culture bottles with a dollop of wet yeast added to the side of the chamber. This allows the flies to reach sexual maturity with a standard, protein-rich diet. Then, two females and one male are transferred into each well of a 48-well plate that has been prepared with the layer of wet yeast on top, as described above, and maintained at 25 °C for a one day “pre-egg-lay” period. This number of flies was chosen because we found that there was lower interwell variability with two females compared to one, while more than three flies per well increased the frequency of wells that needed to be excluded because one or more flies in the well died during the assay period. We included the one-day pre-egg-lay period before starting the time course because it moderately reduced the interwell variability, as indicated by a decrease in the coefficient of variation of the “Day 1” egg counts from 0.75 or 0.83 without the pre-egg-lay period (Figure 1B, “Grape Juice” condition or Figure 2D, “Mixed” condition, respectively) to 0.63 with the pre-egg-lay period (Figure 2E). After the pre-egg-lay period, the flies are transferred to a new 48-well plate every day so they have constant access to a moist, protein-rich diet that exposes them to the desired dietary condition. The wells contain either base media with yeast mixed in, as described above, on days in which fecundity will be measured (e.g., days 1, 3, and 7 in Figure 2E) or with yeast on top on the remaining days (e.g., days 2, 4, 5, and 6 in Figure 2E). We found that the strategy resulted in a consistent rate of egg laying for at least 3 days, with only a modest drop off by 7 days (Figure 2B–E). Specifically, we observed an average of 35.9, 42.7, and 19.8 eggs per well after 1, 3, and 7 days, respectively. While this is lower than the rates of egg laying that are typically observed when flies are maintained in vials [34], it is a high enough baseline that experimentally induced effects on egg-laying rates should be detectable. Indeed, as we demonstrate below, this assay was capable of detecting dose-dependent effects of exposure to two different chemicals. Using 1 percent agarose rather than 2 percent did not improve the rates of egg laying, suggesting that the stiffness of the media is not the cause of the reduced rates of egg laying over a 7-day time course (Appendix A). Thus, we find this protocol to be a useful compromise that balances physiological needs with accuracy and efficiency.

### 3.2. Fly Transfer Device

The 48-well plate format is useful for testing multiple different experimental conditions in a high-throughput manner. However, the plastic lid does not fit tightly enough over the surface of the wells to ensure that flies cannot escape or move between wells, and it is very difficult to anesthetize and transfer the flies to a new plate without losing some or mixing them up. In addition, manually transferring each fly to the new well is a time-consuming process. To overcome these challenges, we designed a custom transfer device with tapered cylinders that fit snugly into each well of a 48-well plate (Figure 3). The wide ends of the cylinders are blocked off with a fine mesh that allows for air exchange while preventing the flies from escaping. To transfer the flies to a new 48-well plate, the flies are anesthetized by inverting the transfer device and plate onto a CO_2_ pad so that the flies fall onto the mesh surface and the old plate can be replaced with a new one. The transfer device and new plate are removed from the CO_2_ pad and, when the flies recover, the transfer device and plate are inverted so that the plate is upright again (Appendix A). This process is efficient, reducing the transfer time from approximately 5 min to less than 30 s, and ensures that flies are accurately transferred to the new wells and are unable to escape during the incubation periods. An additional benefit is that flies are only exposed to CO_2_ for a very brief period of time (less than a minute) so the potential effects that CO_2_ may have on the health or fecundity of the flies is minimized [47]. 

### 3.3. RoboCam Platform

Our next goal was to develop a high-throughput method to image the surface of each well. We first experimented with different forms of illumination and found that lighting from underneath using an ultra-thin LED Light Tracing Box was ideal, as it minimized shadows and reflections from the plastic and provided a good contrast between the eggs and the media. Then, we developed the RoboCam platform to automate the image acquisition. This platform integrates the precise robotic functionality of a 3D printer with the computational efficiency of Raspberry Pi computers and a high-resolution digital camera with a telephoto lens into a programmable robotic camera (Figure 4A–G and Table 1). An external hard drive connected to the Raspberry Pi ensures safe storage of the images. We developed a Graphical User Interface (GUI) in Python to control the system (Appendix A). Through this interface, the Raspberry Pi issues serial commands to control the RoboCam, facilitating precise movements in a snake path across the plate to sequentially position the camera over each well (Figure 4H), capture images, and store the image files on the external hard drive. Before each run, the RoboCam is calibrated to the well plate by aligning a circular target with the wells in each of the four corners of the plate, which is stored as a CSV file along with the well scan pattern. Focus is manually set on the lens or by moving the camera in the z direction through the GUI. Upon completion of the run, the images stored on the external hard drive are transferred to another computer for the image analysis steps. The advantages of using the RoboCam platform to collect images are discussed in detail in the Discussion section. Scientifically, the main benefit of the RoboCam is the consistency of the images. In all the images taken by the RoboCam, the wells are in the exact same position, greatly facilitating the data analysis. On the technical side, the modular set-up of the RoboCam allows changes to its design, so that scientists can modify it depending on the specific purpose of their research. 

https://store.anycubic.com/products/anycubic-i3-mega (accessed on 6 August 2024).https://www.amazon.com/gp/product/B0755C2CBF?th=1 (accessed on 6 August 2024).https://www.arducam.com/product/12-3mp-477m-hq-camera-module-for-raspberry-pi-with-135d-m12-wide-angle-lens/ (accessed on 6 August 2024).https://www.adafruit.com/product/4562 (accessed on 6 August 2024).https://www.canakit.com/raspberry-pi-3-model-b.html?cid=usd&src=raspberrypi (accessed on 6 August 2024).

### 3.4. Image Segmentation

With the steps described above in place, we next sought to develop an image segmentation pipeline that would automatically identify and quantify eggs from the images taken by the RoboCam. First, we tested Quantifly [40], which utilizes a density estimation approach to quantify the number of dense objects in the image. To determine the accuracy of this approach, we carefully hand-counted the number of eggs in 133 images and compared these manual counts to the results from Quantifly using multiple different sigma values. The best performance we could obtain from Quantifly on this dataset was an R^2^ of 0.74 (Figure 5A,D). Next, we investigated whether Stardist, a deep learning method that is based on star–convex shape detection [45,46], could achieve better performance. Indeed, the pretrained Stardist model, 2D_versatile_fluo, achieved an R^2^ of 0.79 with the same set of images (Figure 5B,D). To improve upon these results, we generated a custom model (flyModel2) trained on a set of images with hand-drawn outlines around each egg as ground-truth annotations. This increased the accuracy to an R^2^ of 0.91. We examined the sources of the errors in this approach and noticed that, in some cases, eggs near the edge of the well were reflected in the clear plastic wall and the reflections were counted as separate eggs (Figure 5E). Increasing the sensitivity threshold reduced the number of these false positives but also increased the number of false negatives. Thus, to eliminate this type of error, we used an ellipse detection tool to identify the well boundary and restrict the analysis to the portion of the image within this boundary (Figure 5F). The RoboCam positions each well in precisely the same location in each image, so we were able to accurately map the well in all the images from the same plate by performing the well detection on the first well and then using the coordinates to position the mask thereafter (Figure 5G). Together, these strategies comprise a highly accurate (R2 = 0.98) and efficient pipeline for quantifying egg number (Figure 5C,D).

### 3.5. Quantification of the Effects of Chemical Exposure

To evaluate the performance of this pipeline in experimental contexts that may alter fecundity, we first used it to quantify the rates of egg laying in two additional wildtype strains, OregonR and CantonS. We observed similar rates of egg laying over a 7-day time course in all three strains (Appendix A). Next, we assayed for changes in the rate of egg laying upon exposure to either rapamycin, which is a potent inhibitor of Tor signaling [35], or bendiocarb, which is an insecticide used to control mosquito populations [48] (Figure 6A). Rapamycin has been shown to significantly reduce *Drosophila* fecundity at concentrations ranging from 30 to 400 μM in the culture media [35,49] and, though the impact of bendiocarb exposure on *Drosophila* fecundity is not known, several studies have assessed viability. Specifically, culturing wildtype flies in glass vials coated with 0.1 μg bendiocarb has minimal impact on viability [50], whereas 24 h of exposure to a 313 μM bendiocarb along with a liquid sucrose diet substantially reduces viability [51]. Using these results as a guide, we tested fecundity following 1–7 days of culture with media that contained either 0.1% DMSO alone, which does not impact fecundity [52], or 0.1% DMSO with 0.1 μM, 10 μM, or 25 μM of the chemical. We adapted our standard workflow (Figure 2A) for this purpose so that flies were exposed to fresh chemical every day of the 7-day time course (Figure 6A). Because the chemical is dissolved in the media, the primary route of exposure is through dietary intake of food and water, though other routes of exposure, such as through the cuticle, may also occur. The strategy of adding the chemical directly into the media has the advantage that any potentially aversive olfactory or gustatory cues from the chemical will be offset by the highly motivating need for food and water. Assuming that adult flies consume approximately 1.5 μL of food and water per day [53], these doses correspond to a dietary intake of 0.14–34.3 ng of rapamycin and 0.03–8.3 ng of bendiocarb per day.

For both chemicals, we observed a progressive decrease in the number of eggs with increased dose and length of exposure (Figure 6B–F). However, these observations are based on quantification of eggs from at least 60 wells per condition. To assess whether the effects of rapamycin and bendiocarb in the diet could have been detected with fewer repeats of each condition, we downsampled the data by randomly choosing 4, 8, 16, or 24 wells from each condition (day and chemical dose) 100 times and performing pairwise *t*-tests on each of these randomly selected subsets of the data (Figure 7). We found that nearly all of tests with 16 or 24 wells sampled from the Day 3 and Day 7 datasets reproduce the conclusions about statistical significance from the entire dataset (except for 0.1 µM bendiocarb at Day 7) and that most tests with eight wells sampled were able to detect a statistically significant difference in fecundity at the highest (25 µM) dose of rapamycin and bendiocarb at Day 3 (Figure 7). Thus, this pipeline is able to efficiently detect dose- and time-dependent experimentally induced decreases in fecundity.

## 4. Discussion

Here, we present a high-throughput pipeline for measuring *Drosophila* fecundity that links together multiple innovations. First, the strategy we developed for maintaining robust egg laying in conditions that are compatible with the 48-well format and high-quality imaging of the eggs creates a new opportunity for efficiently testing many different dietary conditions or chemical exposures at a time. The 48-well format substantially increases the efficiency of culturing the flies in different conditions and significantly reduces the cost of media and chemicals since the volume used for each well is much lower than the volumes used in standard culture vials. Next, we coupled this with custom devices that are low-cost and open source, and we engineered the RoboCam. By integrating consumer-grade 3D printers, cameras, imaging chips, and Raspberry Pi computers, the RoboCam allows the acquisition of high-resolution images in a consistent and accurate way. This utilization of readily available consumer products delivers high precision, reproducibility, and efficiency at low cost. We implemented a GUI to be used with the Robocam, which provides a user-friendly interface for image collection. In addition, because the RoboCam is a modular platform, it is possible to change the different parts for desired functions, allowing for selecting the most suitable camera and lens for different applications and optimizing performance across various tasks. In addition, the 3D printer within the RoboCam platform retains its original printing function, thus maintaining its dual-purpose capability. From a technical point of view, the protocol’s dedicated operation on the Raspberry Pi ensures the absence of unexpected system updates and the need for a network connection, leading to a more stable and secure operation. Designed for extended use, once that the G-code instructions are programmed through the GUI, the RoboCam can run and record data without human oversight; the recorded data are stored in an external hard drive, which allows for efficient data transfer to other computers. This arrangement also supports the simultaneous and independent operation of multiple RoboCams, so it can be scaled as needed. Thus, a major advantage of the Robocam is that it yields more reproducible, accurate, and consistent images than can be easily obtained with manual methods. Lastly, we have developed an image segmentation process that combines a deep learning model with additional image segmentation tools to accurately quantify the number of eggs in each image. 

Oogenesis in *Drosophila* is highly conserved, making it a useful model for studies of reproductive health [9]. Indeed, 69–78% of genes involved in *Drosophila* female reproduction have vertebrate orthologs [54], and some aspects, such as meiosis, are particularly well-conserved at both the gene and mechanistic level [55]. The development of this high-throughput method to measure *Drosophila* fecundity introduces a new tool in the toolbox for these studies. This method is well-suited for toxicology screens to identify compounds in the environment that impact fertility [56,57] and thus aligns well with the need for new approach methodologies (NAMs) that are being adopted by regulatory agencies worldwide [2]. Likewise, this method would also be useful for broader toxicology and ecotoxicology [58] screens, as well as for screens to identify small molecules that may be useful for therapeutics [59,60] and insect population control [61]. In addition, since oogenesis is highly dependent on and coupled with other organ systems in the body, a high-throughput fecundity screen may be useful for identifying small molecules, dietary conditions, or other experimental variables such as age that affect other organs. For example, the rate of oogenesis is regulated by signals from the brain and fat body, so changes in fecundity could be used as an assay for RNAi or CRISPR screens that disrupts gene expression in these organs. Moreover, a high-throughput assay for fecundity would be an efficient way to characterize large collections of natural variants [62,63]. Lastly, although we have focused on fecundity here, this format may also be an effective way to rapidly screen for experimental conditions that affect embryo hatching rates or adult viability. Thus, we expect that this new pipeline will have a wide range of applications.

However, because ovarian function depends on other aspects of physiology, an experimental condition that impacts fertility may be operating through any one or more of multiple different mechanisms, and follow-up experiments would most likely be required to determine whether the observed effects are due to an impact on a conserved aspect of oogenesis, such as meiosis, or not. A limitation of the high-throughput method is that the rate of egg laying in the 48-well plates is lower than it is when flies are maintained in standard conditions, perhaps because factors such as the confined space of the wells or the composition of the media discourage egg laying, so the assay is not measuring oogenesis at maximum capacity. In addition, there is substantial variability in the number of eggs laid in each well, which reduces the sensitivity of the assay. This may be exacerbated if a test chemical has an aversive effect that alters feeding patterns. However, we were able to partially counterbalance the variability between individual flies by adding two females to each well and the variability between wells by repeating the same condition in multiple wells. With this approach, the impact of the chemicals we tested was easily detectable, and accepting these limitations in exchange for the ability to screen fecundity in a low-cost, high-throughput manner will likely be a worthwhile tradeoff in many cases. Moreover, our analysis of the downsampled data (Figure 7) suggests that similar conclusions can be reached using many fewer wells per condition than we used here. This may be advantageous in situations, such as a large-scale screen, where increased efficiency at the expense of some loss of accuracy is acceptable. Taken together, we expect that this new pipeline will have a wide range of applications in *Drosophila* research.

## Figures and Tables

**Figure 1 toxics-12-00658-f001:**
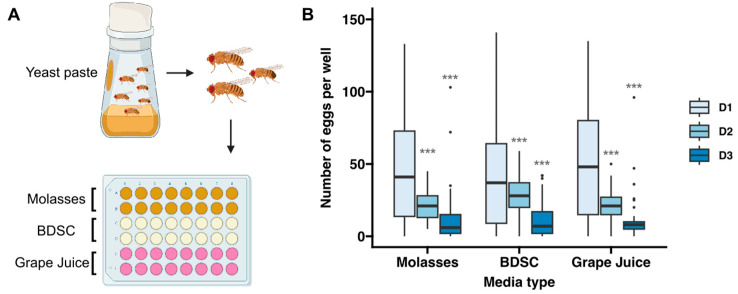
Quantification of fecundity over three days on different types of media. (**A**) Diagram showing the workflow for this assay. Flies were fed wet yeast paste for two consecutive days and then 2 females and 1 male were transferred into each well of a 48-well plate with molasses-based, BDSC-based (tan), grape juice–agarose-based media (16 wells for each condition, as indicated in gold, tan, and pink colors, respectively) and allowed to lay eggs for 23 h. (**B**) Graph showing the number of eggs laid in each condition, as determined by manual counts. Sample sizes per condition range from 29 to 61 wells and the data were collected from 2–4 independent replicates. Asterisks indicate statistical significance using Bonferroni-corrected pairwise *t*-tests between D1 and D2 or D3 for each condition. *** *p* < 0.001.

**Figure 2 toxics-12-00658-f002:**
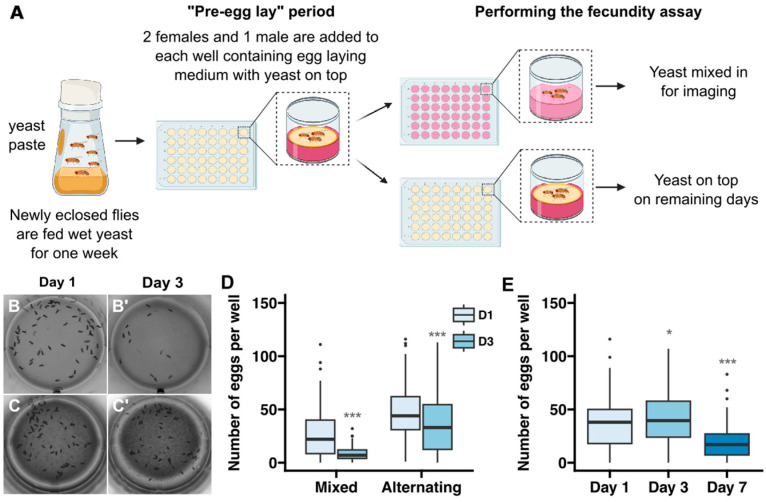
Development of protocol for assaying fecundity over 7 days. (**A**) Diagram showing a workflow in which flies are maintained on media with yeast mixed in on the days in which images will be acquired to assess fecundity and on media with yeast on top during the remaining days. In the “alternating” conditions shown in panels (**C**–**E**), flies were put on media with yeast mixed in on days 1, 3, and 7 and on media with yeast on top on days 2, 4, 5, and 6. (**B**–**D**) Representative examples of wells from a 3 day time course in which flies were maintained in wells with the yeast mixed into the media at all time points (**B**) or alternating between wells with yeast mixed into the media on days 1 and 3 and yeast on top on day 2 (**C**) and a graph showing the number of eggs laid in each regime (**D**). (**E**) Graph showing the number of eggs laid in the alternating regime over a 7-day time course. Sample sizes in (**D**,**E**) per condition range from 78 to 144 wells and the data were collected from 3–7 independent replicates. Asterisks indicate statistical significance using Bonferroni-corrected pairwise *t*-tests between D1 and D3 for each condition in (**D**) and D1 and D3 or D7 in (**E**). * *p* < 0.05, *** *p* < 0.001.

**Figure 3 toxics-12-00658-f003:**
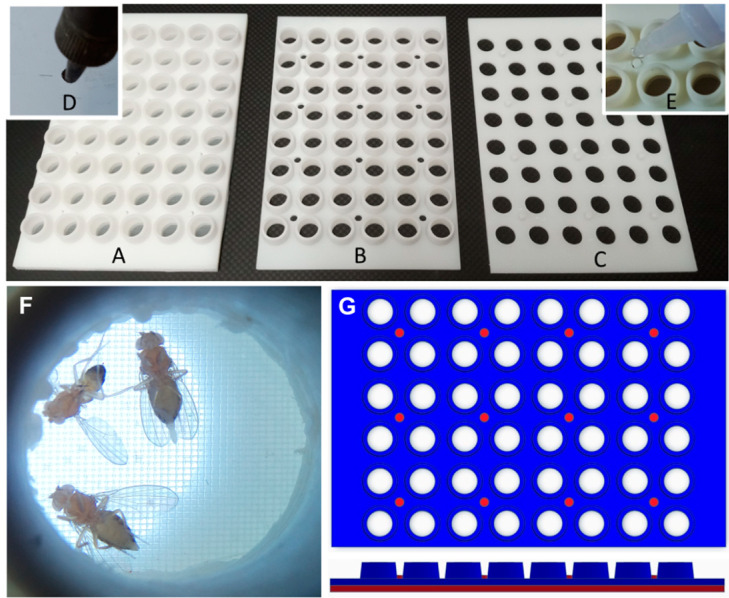
Fly transfer device. (**A**–**G**) A fly transfer lid (**A**) is constructed from a layer of nylon mesh bonded between an array of cups (**B**) and holes (**C**). Studs in the hole layer (**C**) pass through corresponding holes in the cup (**B**) and nylon layers, creating individual gas exchange tops for each well. A hot soldering iron, guided by holes in a metal template, creates holes in the nylon mesh (**D**). The final assembly (**E**) studs are bonded with cyanoacrylate glue. (**F**) When placed on a CO_2_ plate, anesthetized flies fall into their corresponding cup, enabling the plate to be replaced and flipped, simultaneously transferring all flies into new wells. (**G**) Top and side view of the fly transfer device. The cup layer is shown in blue and the hole layer is shown in red.

**Figure 4 toxics-12-00658-f004:**
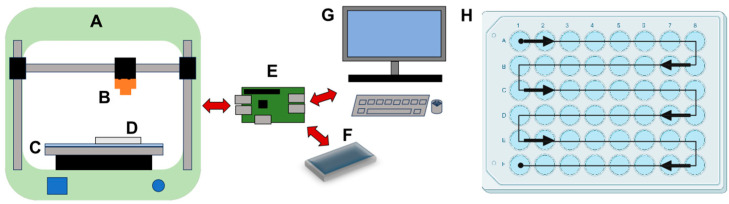
RoboCam device for automated image capture. A 3D printer (**A**) is modified by adding a camera (**B)** and light plate (**C**), where the 48-well plate (**D**) is located. The light ensures good contrast between the eggs and the media while minimizing shadows and reflections. A single-board computer (**E**) controls the x, y, and z movement of the camera (**B**) and saves captured images on a hard drive (**F**). The user programs the RoboCam using a graphical user interface with a keyboard, mouse, and monitor (**G**). (**H**) Camera system moves in a snake path, centering over individual wells.

**Figure 5 toxics-12-00658-f005:**
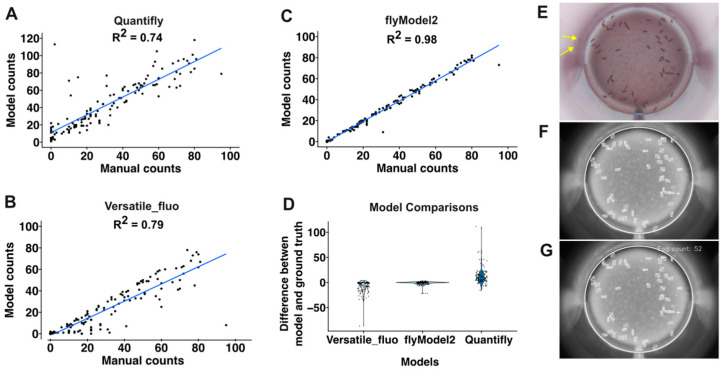
Automated image analysis pipeline. (**A**–**C**) Graphs showing comparisons of manual egg counts to automated egg counts using Quantifly (**A**), Stardist with the Versatile_fluo model (**B**), or Stardist using a custom-build model, flyModel2 (**C**). Each dot shows the manual and automated counts from a single well. (**D**) Graph showing the error between manual and automated egg counts. Each dot is the difference between the automated count and the manual count (ground truth). (**E**–**G**) Images showing individual steps in the automated image analysis pipeline. Starting from the raw image (**E**), the Stardist model identifies the eggs and the elipse detection tool identifies the well edge (boxed regions and white circle, respectively in Panel **F**). Then, the number of boxed regions (eggs) within the well are counted (**G**). Eggs reflected in the well walls are indicated with yellow arrows.

**Figure 6 toxics-12-00658-f006:**
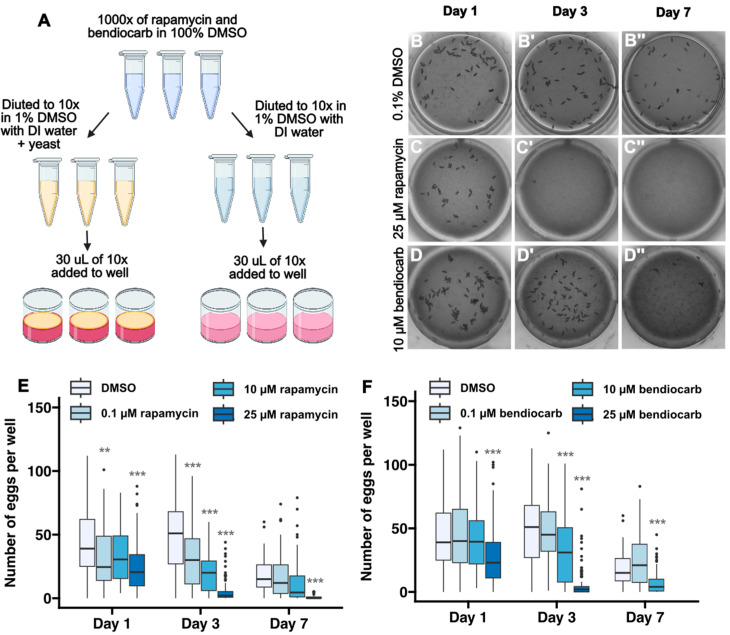
Detection of the impact of dietary exposure to rapamycin and bendiocarb. (**A**) Diagram showing workflow for fecundity assays in which chemicals are added to the diet. Chemicals are dissolved in 100% DMSO to a 1000× concentration. Then, they are diluted to 10× concentration in 1% DMSO with either water or water plus yeast. Finally, 30 µL of the 10× solution is pipetted into each well, which contains 300 µL of media, producing a final concentration of 1x chemical in 0.1% DMSO. Using this protocol, flies were exposed to 0.1% DMSO or indicated concentrations of rapamycin or bendiocarb in 0.1% DMSO for 7 days. Egg counts were quantified on days 1, 3, and 7. (**B**–**D**) Images of wells at 1, 3 or 7 days after exposure to 0.1% DMSO (**B**), 25 µM rapamycin (**C**), or 10 µM bendiocarb (**D**). (**E**,**F**) Graphs showing the number of eggs laid in the indicated conditions over the 7-day time course. Flies did not survive for 7 days on 25 µM bendiocarb. Asterisks indicate a statistically significant difference compared to the DMSO condition on the same day ** *p* < 0.01, *** *p* < 0.001 using pairwise t-tests with a Bonferroni multiple hypothesis test correction.

**Figure 7 toxics-12-00658-f007:**
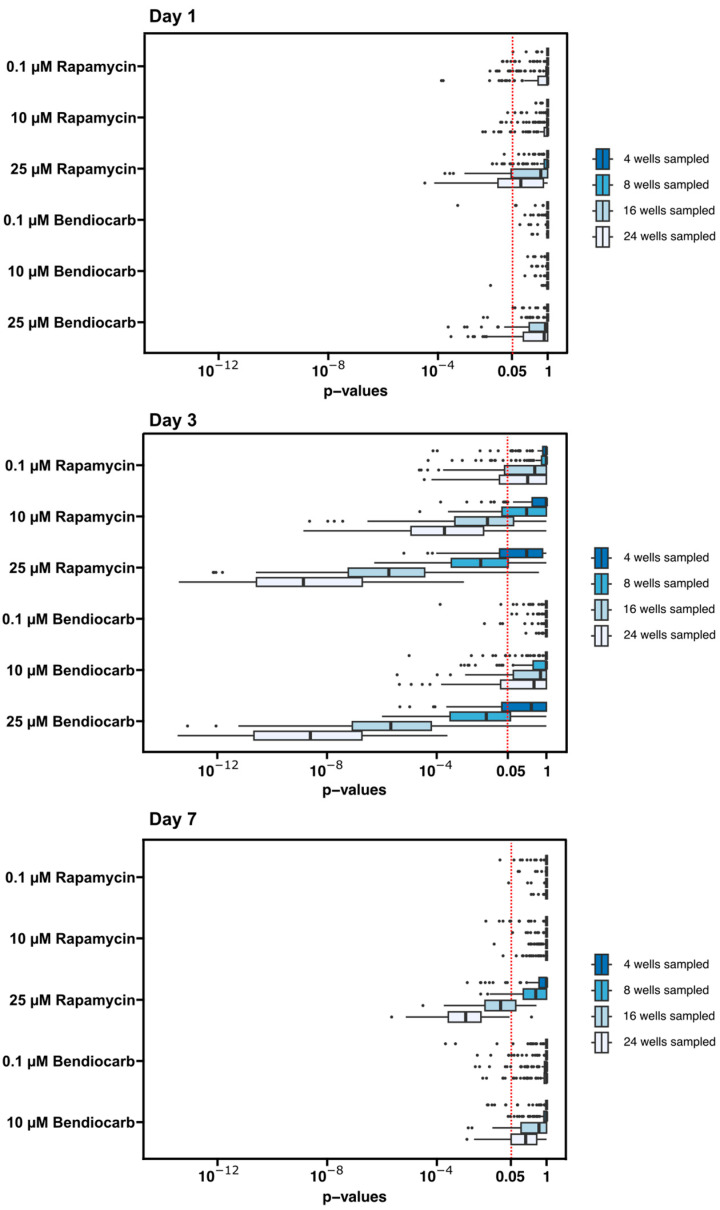
Downsampling of data from dietary exposure to rapamycin and bendiocarb. The *p*-values from 100 iterations of downsampling the data by randomly selecting 4, 8, 16, or 24 wells from the day 1, day 3, and day 7 datasets are displayed on the graphs. *p*-values to the left of the red dotted line are below the conventional 0.05 threshold for statistical significance.

**Table 1 toxics-12-00658-t001:** RoboCam Components.

Website	Model	Cost	Specification
1	AnyCube Mega-S	USD 239	Positioning Accuracy: 12.5 um X/Y, 2 um Z. Build size: 210 × 210 × 205 mm
2	Light Panel	USD 21	USB powered, dimmable, white LEDs
3	Camera	USD 50	Sony IMX477 sensor, 12 Mpix
4	Lens	USD 50	16 mm focal length, F1.4–16, C Mount
5	Raspberry Pi	USD 35	Quad Core, 1.2 GHz, 1 GB RAM

## Data Availability

Data is contained within the article or Appendix A.

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
