# Peer review of "A High-Throughput Method for Quantifying Drosophila Fecundity"

_toxics, 2024, doi:10.3390/toxics12090658_

Round 1

Reviewer 1 Report (Previous Reviewer 1)

Comments and Suggestions for Authors

No additional comments.

Author Response

We thank you for your positive assessment of this work and your feedback, which helped us improve the manuscript.

Reviewer 2 Report (Previous Reviewer 3)

Comments and Suggestions for Authors

The revised manuscript can be accepted in present form.

Author Response

We thank you for your positive assessment of this work and your feedback, which helped us improve the manuscript.

Reviewer 3 Report (New Reviewer)

Comments and Suggestions for Authors

The manuscript by Gomez et al describes a high-throughput method of evaluating fecundity of fruit flies exposed to different environmental toxicants. They describe a custom-built apparatus that enables them to transfer insects quickly and easily into multi-well plates and the development of automated imaging tools to count the number of eggs laid by insects exposed to different xenobiotics.

The manuscript was clearly written, the rationale of using insects for such studies was nicely justified, and the results supported their claims that a high throughput method of counting fly eggs was achievable and could yield meaningful results. I have only a few minor points for the authors to consider:

1.      Figure 3 does not clearly show the assembled fly transfer device. A schematic diagram (side view and top view) would help clarify how it is assembled and used. Line 159 indicates that a Supplemental Movie 1 shows this setup, but it was not included in the package of files to view. I suspect seeing that video would help visualize the apparatus and its assembly, but I still think a schematic would be useful.

2.      Will the instructions for 3D printing of the transfer device be publicly available? I did not see that information on the Nystul lab web page.

3.      Does DMSO impact fly fecundity? How was the 0.1% DMSO dose chosen? Presumably this dose has minimal impact on the flies’ fecundity, but no reference is provided. In instances where the tested compounds are water-soluble, DMSO would presumably not be required.

4.      Aside from counting eggs, could this method allow you to assess mortality of the flies? This could translate to a toxicity test on the adults.

5.      Does the imaging system enable you to visualize the hatch rate of the eggs? If you created a thin enough layer of translucent agar, it might be possible to see larvae, which would provide a measure of fertility (actual number of offspring) vs fecundity (potential of reproduction). Flies can lay non-viable eggs, and it could be more informative to observe the hatch rate than the number of eggs when assessing the impacts of the xenobiotics on the insects. It might be helpful to provide some commentary on this distinction in the Discussion.

Author Response

Reviewer 4 Report (New Reviewer)

Comments and Suggestions for Authors

This manuscript by Gomez, et al. examines a new method for quickly and efficiently quantifying egg production in Drosophila. Briefly, the developed an approach for rearing small numbers of flies in a 48 well plate that is suitable for photographing eggs, they developed a novel method for quickly transferring all flies on the plate to fresh media, and the engineered an automated camera system capable of automatically photographing eggs in the media. Overall, this is an interesting methods paper that addresses a problem with counting eggs from large numbers of flies in an efficient and accurate manner. While I have serious reservations on whether fruit flies, with completely different reproductive biology and a suite of invertebrate specific hormonal cascades, can serve as a model for mammalian reproduction, it can be valuable for a range of Drosophila specific assays as reviewed in the discussion. I have no overall concerns with two components of the counting system, specifically the fly transfer device and the automated photography. Indeed, I found the fly transfer device very clever. However, in regards to the growth media used, the large reduction in egg production is concerning. The authors state that their observed egg production (D1:35.9, D3:42.7 and D7:19.8) is lower than typically observed in flasks. Looking at the provided reference, typical egg-laying on day 3 is ~80 eggs/female, or nearly double what the authors achieved, which is quite large. Clearly there are physiological changes in the fly due to the diet, crowding, or some other stress factor. And while the authors address this weakness in the discussion, it still greatly limits the utility of the assay since these stressors are likely to influence results on downstream bioassays.

Additional concerns:

Figures 1: Were the results statistically significant and what statistical tests were used? Were there significant differences between treatments? What was the sample size? How many independent biological replicates were performed? Why was a wet yeast paste control not included?

Figure 2D and E: Similar concerns as Figure 1.

Figure 5C. Any ideas as to what happened with the two outliers where the models dramatically undercounted?

Lines 450-455. This statement is not shown in figure 6, although the data is in S4. Graphs representing 8, 16, and 24 wells (i.e., the graphs in S4) should be included in Figure 6 or as Figure 7.

Figure S2 and S3. Statistical significance, if any, and sample size should be included.

In summary, this is an interesting methods paper that would be a reasonably good fit for the special issue “Feature Papers in the Novel Methods in Toxicology Research”. The authors need to provide details of the bioassays and statistical analysis, but assuming that sufficient sample sizes were assayed I do not have major concerns with the manuscript.

Author Response

This manuscript is a resubmission of an earlier submission. The following is a list of the peer review reports and author responses from that submission.

Round 1

Reviewer 1 Report

Comments and Suggestions for Authors

Overall, this is an interesting work establishing a new strategy to improve and automatize egg-laying quantifications in Drosophila melanogaster. The authors show that using a 3D-printed device, adult flies can be maintained in a 24-well plate with an egg-laying media that enables collect the Drosophila embryos in a specific time window. Interestingly, the 3D-printed device allows the flies to be anesthetized and transferred to a new 24-well plate containing fresh egg-laying media to quantify the following embryo collection at a different time. Moreover, the authors show that by adapting a camera to a 3D printer and incorporating such a system into a computer, embryos collected in each well from the 24-well plate can be automatically imaged and quantified. This innovative strategy provides the researcher with a straightforward way to evaluate Drosophila egg-laying counts on a surface exposed to components that may impact Drosophila fecundity.  While the authors show that their strategy is suitable for assessing Drosophila egg laying on a surface exposed to two chemicals (Rapamycin and bendiocarb), some aspects need to be addressed to improve the paper.  

The introduction reads well. The authors state the need for an automatized strategy for accurately quantifying egg laying in Drosophila. They also propose their strategy as suitable for assessing the impact of prolonged chemical exposure on fecundity.  

However, given that most of the work focuses on the generation and construction of the 3D-printed device used and the automatization of the imagining and subsequent analysis of such images, the material and methods section could be more detailed to support and ensure appropriate reproducibility by other researchers. 

My first recommendation is that the supplementary Figure S1 needs details about the parts that could be more obvious for a person unfamiliar with using a 3D printer routinary. This section needs to have enough detail to permit the appropriate ensemble of the RoboCam. I recommend that the supplementary figure and the supplementary figure legend be improved and indicate which parts of the 3D printer already come with the printer and the parts that the researcher needs to ensemble. An option could be adding arrows or numbers and the names of the parts that need to be assembled for the 3D printer, as in the schematic representation in Figure 4. Are the lightbox and the digital camera the only components that need to be attached to the 3D printer? Is there any adapter required to couple the camera to the printer?

The second concern is that references after #38 need to be included in the reference section in the text. The authors need an appropriate literature citation to address the prolonged exposure of Rapamycin and Bendiocarb.

For example, in section “3.5 Quantification of the effects of chemical exposure,” in lines 409 and 410, the authors cite reference 42 to refer to rapamycin as a potent inhibitor of Tor signaling that reduces fecundity in Drosophila. In addition, in lines 411 and 412, the authors use reference #43 to mention that bendiocarb is an insecticide for the mosquito population. 

I did not find such references in the manuscript. However, I found a preprint version of the manuscript and realized that References 42 and 43 are unrelated to the text’s sentence (Lines 409 to 412). Reference 42, “Chemical cues that guide female reproduction in Drosophila melanogaster,” is a review that highlights the chemical signals in reproduction, including pheromones, and how they mediate attraction and general reproductive biology in Drosophila. Reference 43, “Environmental influences on reproductive health, the importance of chemical exposure,” reviews the effects of prolonged chemical exposure on reproduction, including some pesticides. Although the topic is related, such references are not directly related to the doses used in the work for any chemicals (Rapamycin and Bendiocarp) or the effects mentioned in the text. There are currently other works in Drosophila that previously have shown that Rapamycin affects fecundity and life span using doses between 50mM to 200-400mM; such work may be helpful to improve the text and discussion. In addition, has Bendiocarp been used before in Drosophila

Given that the proposed strategy is thought to be applied as a strategy to measure the effect of prolonged chemical exposure, the route of exposure needs to be discussed. Lines 499 -502 mention limitations, including the variability in the number of eggs due to the number of flies. The route of chemical exposure may also be discussed as a source of variability in egg-laying. It is biologically relevant because, for example, does Bendiocarp act as a repellent in mosquitoes? It is possible that Bendiocarb can act as a repellent on the surface of the egg-laying medium, and such fecundity effects may be due to a repellence effect. Such discussion might improve the limitations and interpretation of the impact on fecundity under prolonged chemical exposure. 

 Other minor comments.

Drosophila in italics

Line 115 and line 246. If the authors refer to Bloomington Drosophila Stock Center, the abbreviation “BIDC” must be corrected. If authors are not referring to the BDSC, please add an abbreviation meaning.

Reviewer 2 Report

Comments and Suggestions for Authors

Please find in the attached file my comments.

Reviewer 3 Report

Comments and Suggestions for Authors

In this manuscript, Gomez and colleagues tested and developed a high-throughout method for quantifying fruit fly Drosophila melanogaster fecundity. The authors have carried out appropriate experiments, and explained the rationale and results well step by step in the main text. I like this work, and agree with the authors that it will be useful for Drosophila research. The manuscript can be accepted after italicising the word "Drosophila" throughout the manuscript. 

Reviewer 4 Report

Comments and Suggestions for Authors

The manuscript presents a novel and comprehensive high-throughput method for quantifying Drosophila fecundity. The authors have developed a multifaceted approach combining a new multiwell fly culture strategy, a 3D-printed fly transfer device, a low-cost robotic camera (RoboCam), and an image segmentation pipeline. This method is supposed to overcome the limitations of traditional fecundity assays, which are typically labor-intensive and difficult to perform in a high-throughput manner. By automating the transfer of flies and the imaging and quantification of eggs, the proposed system allows for efficient and accurate assessment of fecundity under various experimental conditions. 

Despite the high quality of the manuscript, there is a critical aspect that needs to be addressed before it can be approved for publication (unless I failed to identify these in the manuscript). In my opinion, the authors have two major alternatives. They should either present the identification and characterization of new toxic compounds whose impact of fly oviposition had never been described and would be of interest for any specific reason, or make available all the necessary data for any person to implement their strategy. For that, the authors must make available all the 3D print models, codes for the robotic machinery, the imaging acquisition, and processing pipelines. This transparency is crucial for reproducibility and further validation by other researchers in the field.

General Comments:

The English quality of the manuscript is excellent, ensuring clarity and readability.

The figures are well-made and highly descriptive, effectively supporting the presented data.

The legends are well-crafted, providing the necessary context and details for understanding the figures.

Other Points for Improvement:

While the description of Drosophila biology is extensive (perhaps excessively), the authors should clarify why the utilization of flies as a model for toxicology and fertility is important. To strengthen their argument, they should incorporate references from the literature that validate their claim.

Consistency in tense usage (preferably past tense) is recommended for the description in the Materials and Methods section.

The manuscript should address the dynamic of oviposition across different days. Due to their methodology, the authors are missing data points at different days across the period of the experiment. Currently, it is not clear why the loss of this data does not impact the sensitivity of the assay. Clarification is needed on how this missing data affects the overall results and whether the assay remains sufficiently accurate for this type of investigation.

The manuscript could clarify how this pipeline compares against other assays in terms of sensitivity and accuracy (for example, their own previous methods and estimates). The authors could demonstrate whether this method provides a significant improvement over existing methods for analyzing this type of data and conducting these experiments.

A supplemental movie demonstrating the transfer device and the process of transferring flies would be beneficial.

The manuscript should address to what extent this method is genuinely high throughput. It appears that a large number of plates may be required for these assays, necessitating significant manual preparation and handling. The authors should clarify whether this method allows for the testing of hundreds of chemicals in terms of their toxicological effects on oviposition (high throughput). This might require a fully automated and scalable protocol beyond what is currently presented.